# C9orf72 arginine-rich dipeptide repeats inhibit UPF1-mediated RNA decay via translational repression

Yu Sun[1,2], Aziz Eshov[1,2], Jeffrey Zhou[1,2], Atagun U. Isiktas [1,2] & Junjie U. Guo [1,2✉]

Expansion of an intronic (GGGGCC)$_n$ repeat region within the *C9orf72* gene is a main cause of familial amyotrophic lateral sclerosis and frontotemporal dementia (c9ALS/FTD). A hallmark of c9ALS/FTD is the accumulation of misprocessed RNAs, which are often targets of cellular RNA surveillance. Here, we show that RNA decay mechanisms involving upstream frameshift 1 (UPF1), including nonsense-mediated decay (NMD), are inhibited in c9ALS/FTD brains and in cultured cells expressing either of two arginine-rich dipeptide repeats (R-DPRs), poly(GR) and poly(PR). Mechanistically, although R-DPRs cause the recruitment of UPF1 to stress granules, stress granule formation is independent of NMD inhibition. Instead, NMD inhibition is primarily a result from global translational repression caused by R-DPRs. Overexpression of UPF1, but none of its NMD-deficient mutants, enhanced the survival of neurons treated by R-DPRs, suggesting that R-DPRs cause neurotoxicity in part by inhibiting cellular RNA surveillance.

[1] Department of Neuroscience, Yale University School of Medicine, New Haven, CT 06520, USA. [2] Program in Cellular Neuroscience, Neurodegeneration and Repair, Yale University School of Medicine, New Haven, CT 06520, USA. ✉email: junjie.guo@yale.edu

Since the identification of a GGGGCC ($G_4C_2$) repeat expansion within the first intron of *C9orf72* as the major cause of both familial ALS and FTD[1,2], a variety of mechanisms, including haploinsufficiency[3], RNA toxicity[4], and dipeptide repeat (DPR) toxicity[5–7], have been proposed to explain the pathogenicity of this autosomal dominant mutation. The expanded $G_4C_2$ repeat region is transcribed in both directions, producing the C9orf72 pre-mRNA, which contains intronic $G_4C_2$ repeats, and the antisense RNA that contains $G_2C_4$ repeats[8]. Both RNAs accumulate in nuclear foci and, after being exported, can be translated into distinct sets of DPR-containing polypeptides[8]. When overexpressed, the two arginine-rich DPRs (R-DPRs), poly (GR) and poly(PR), have been shown to cause cell death both in vitro and in vivo through a variety of potential mechanisms, such as nucleolar dysfunction, nucleocytoplasmic transport defects, changes in stress granule dynamics, and translation inhibition.

In addition to *C9orf72*, genes encoding RNA-binding proteins such as FUS (fused in sarcoma) and TDP-43 (TAR DNA-binding protein 43) are also enriched in ALS/FTD-associated mutations, suggesting that RNA misprocessing may be a converging point in ALS/FTD pathophysiology[9,10]. Indeed, widespread RNA processing defects have been described by previous high-throughput RNA sequencing (RNA-seq) studies using post-mortem brain tissues from sporadic ALS (sALS) and c9ALS/FTD subjects[4,11]. In normal cells, misprocessed mRNAs are targeted for degradation by multiple RNA surveillance pathways[12–14], such as nonsense-mediated decay (NMD), no-go decay (NGD), and nonstop-mediated decay (NSD). The accumulation of aberrant RNAs in c9ALS/FTD brains hints at the possibility that one or more mRNA surveillance pathways may be compromised.

Here we show that a main reason for the accumulation of aberrant RNAs in c9ALS/FTD brains is the global defect in UPF1-dependent RNA decay pathways including NMD. In cultured neurons, R-DPRs are sufficient to cause NMD inhibition by reducing global translation. Finally, ectopic expression of UPF1 protects neurons from R-DPR toxicity in an NMD-dependent manner. These results suggest a previously underappreciated role of cellular RNA surveillance in the pathophysiology of c9ALS/FTD.

## Results

**UPF1-mediated RNA decay targets accumulate in c9ALS brains**. We re-examined the post-mortem c9ALS, sALS, and control RNA-seq data[11] from the frontal cortex, which shows both RNA foci and DPR pathology in c9ALS/FTD. As shown previously[11], intron retention events were prevalent among c9ALS subjects, as indicated by an increase in read densities for the majority of introns (Fig. 1a, red, median fold change: 2.1). In contrast, the prevalence of intron retention was overall similar between sALS and control subjects (Fig. 1a, blue, median fold change: 0.87). Intron-retaining mRNAs often contain premature stop codons, which would render them substrates for NMD. To assess whether NMD may be defective in c9ALS/FTD, we next quantified the abundance of regulatory targets of NMD. The overall abundance of a list of putative neuronal NMD targets (orthologs of mRNAs that are upregulated in *Upf2* knockout mouse forebrain[15]) ($N = 275$) was increased in c9ALS (Fig. 1b, left, red), but not in sALS subjects (Fig. 1b, right, red). Similar changes were observed when we used two other independent lists of NMD targets identified in human HeLa cells[16,17] (Fig. 1b, orange ($N = 75$) and yellow ($N = 1271$)), further supporting a global NMD deficit in c9ALS brains. In addition, we found that the mRNAs encoding known NMD factors, including UPF3B, SMG5, and SMG6, were significantly upregulated in c9ALS

subjects (Fig. 1c), consistent with a compensatory feedback upon NMD inhibition[18].

Canonical histone mRNAs lack polyA tails and instead have 3′ end stem-loop structures[19]. Histone mRNA decay requires UPF1, but not all NMD factors[19,20]. Similar to NMD targets, we observed overall accumulation of canonical histone mRNAs in c9ALS (Fig. 1d, left), but not in sALS subjects (Fig. 1d, right). In contrast, noncanonical, polyadenylated histone variant mRNAs were largely unchanged between c9ALS and controls (Supplementary Fig. 1), suggesting that accumulation is specific to the UPF1-dependent, canonical histone mRNAs.

To test whether the observed RNA decay deficits in post-mortem brain may be recapitulated in c9ALS motor neurons, we compared the abundance of NMD targets between in vitro differentiated motor neurons from iPSCs derived from control, c9ALS, and SOD1$^{D90A}$-ALS subjects[21]. Again, we observed higher abundance of NMD targets and histone mRNAs in c9ALS but not SOD1$^{D90A}$-ALS motor neurons when compared to control motor neurons (Supplementary Fig. 2).

Collectively, the accumulation of intron-retaining mRNAs, NMD regulatory targets, mRNAs encoding NMD factors, and canonical histone mRNAs suggest that UPF1-mediated RNA decay mechanisms are broadly inhibited in c9ALS/FTD.

**R-DPRs acutely inhibit NMD in cultured cells**. The *C9orf72* repeat expansion can be transcribed in both directions, producing both sense $G_4C_2$ repeat- and antisense $G_2C_4$ repeat-containing RNAs[8], which can be translated into five DPRs: poly(GA), poly (GP), poly(GR), poly(PA), and poly(PR). To test whether any of these gene products may be sufficient to inhibit NMD, we ectopically expressed each of the two repeat RNAs as part of the 5′ untranslated regions (UTRs) of a GFP transcript, and each of four codon-optimized DPRs as GFP fusion proteins[22] in HEK293 cells, and quantified the expression levels of several UPF1-mediated decay targets that accumulated in c9ALS subjects, including four NMD target mRNAs and a canonical histone mRNA. We did not include the poly(GP) expression construct in our analysis because it could not be validated by Sanger sequencing, presumably due to the exceedingly high structure-forming potential of $(GGNCCN)_n$ sequence. DPR-GFP fusion proteins were expressed at similar levels (Supplementary Fig. 3). Within 24 h after transfection, each of the two arginine-rich DPRs (R-DPRs), poly(GR) and poly(PR), increased the abundance of all five tested UPF1 targets (Fig. 2a), whereas no significant changes in target abundance were detected in cells expressing either of the two repeat RNAs nor the two alanine-rich DPRs (Fig. 2a).

To confirm that the selected transcripts are NMD targets in neurons, we treated induced human neurons (iNeurons) with each of two distinct NMD antagonists: caffeine, which non-specifically inhibits UPF1 phosphorylation by SMG1, and cycloheximide (CHX), which inhibits translation elongation. After 24 h, both caffeine and CHX treatments caused significant increases in abundance of the selected transcripts in iNeurons (Supplementary Fig. 4), suggesting that these transcripts are indeed regulated by NMD in human neurons.

To validate the effect of R-DPRs on NMD on a transcriptome scale, we re-examined RNA-seq data from K562 leukemia cells treated with synthetic PR$_{20}$ peptides[23]. We observed substantial accumulation of both NMD targets (Fig. 2b, red, orange, and yellow) and canonical histone mRNAs (Fig. 2b, purple). In addition, multiple NMD factor mRNAs were upregulated in PR$_{20}$-treated K562 cells (Fig. 2c).

To examine the effect of R-DPRs in cell types that are more relevant to FTD, we treated mouse primary cortical neurons with

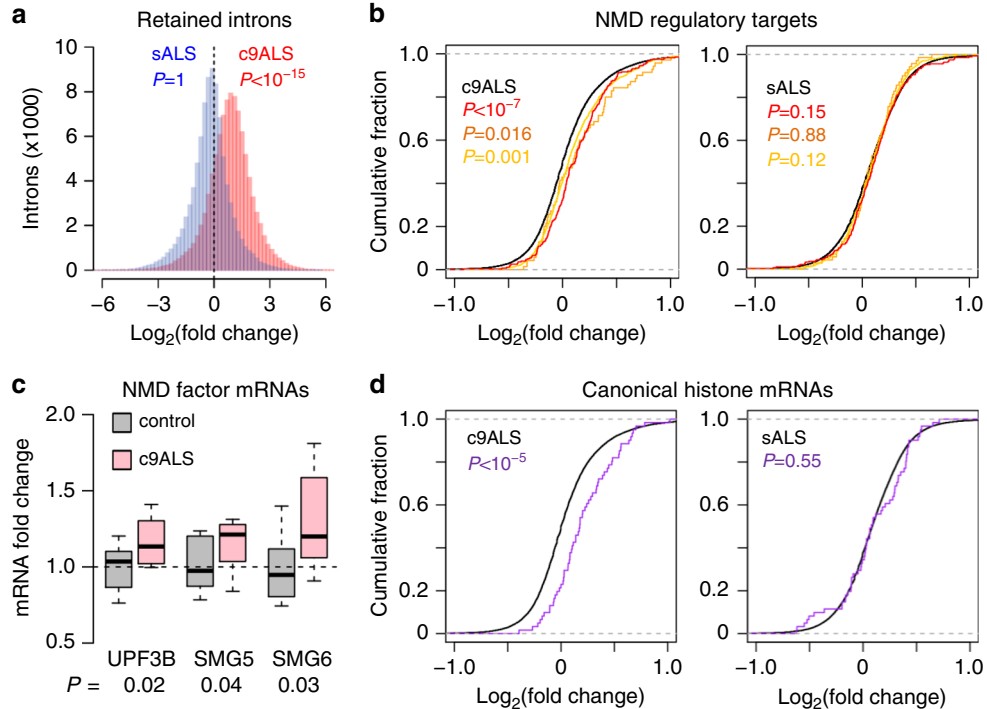

**Fig. 1 NMD targets and canonical histone mRNAs accumulate in c9ALS brains. a** Histograms showing changes in intron read densities between c9ALS and controls (red), or between sALS and controls (blue). $P$ values, one-sided Mann–Whitney $U$ tests. **b** Cumulative distribution functions (CDFs) of changes in RNA abundance for all genes (black), NMD targets identified in Johnson et al. ($N = 275$, red), Tani et al. ($N = 76$, orange), and in Colombo et al. ($N = 1271$, yellow), comparing between c9ALS and controls (left) or sALS and controls (right). $P$ values, two-sided Mann–Whitney $U$ tests. **c** Changes in mRNA abundance of known NMD factors between c9ALS and controls. Expression levels were normalized to the means of control subjects. Boxes indicate the medians and interquartile ranges (IQRs). Whiskers represent 1.5x IQR. $P$ values, two-sided unpaired $t$ tests. **d** CDFs of changes in RNA abundance for all genes (black) and canonical histone genes ($N = 87$, red and blue), comparing between c9ALS and controls (left) or sALS and controls (right). $P$ values, two-sided Mann–Whitney $U$ tests.

synthetic $PR_{20}$ peptides, which are readily taken up by cells and cause neuronal cell death[23]. Within 24 h, $PR_{20}$ caused the accumulation of mouse mRNAs orthologous to the tested human UPF1 targets (Fig. 2d). Similarly, 24 h of $PR_{20}$ treatment also caused the accumulation of UPF1 targets in human iNeurons (Fig. 2e). Collectively, our results suggest that R-DPRs are sufficient to inhibit NMD in both neuronal and non-neuronal cell types.

**R-DPRs cause the recruitment of UPF1 to stress granules.** Previous studies have linked the cellular toxicity of R-DPRs to their influence on stress granule dynamics[24–26]. To test whether NMD inhibition may be linked to stress granule formation, we examined the localization of UPF1 and G3BP1, a stress granule marker, in HeLa cells expressing each repeat RNA or DPR. In control cells, UPF1 exhibited diffuse cytoplasmic localization, with G3BP1-negative small foci that resembled P-bodies (Fig. 3a). In contrast to previous studies[27,28], we did not observe a significant increase in stress granules in cells expressing either $G_4C_2$ or $G_2C_4$ repeat RNAs (Fig. 3a, b), suggesting that stress granule formation may not be a direct consequence of repeat RNA expression. In contrast, poly(GR) and poly(PR) induced the concentration of UPF1 in G3BP1-positive stress granules in 45 and 24% of transfected cells, respectively (Fig. 3a, b). To confirm that UPF1 is a constitutive component of stress granules, we induced stress granules by applying oxidative stress using sodium arsenite ($NaAsO_2$). Indeed, UPF1 and, to a lesser extent, UPF3B, were both recruited to arsenite-induced stress granules (Supplementary Fig. 5). These results are consistent with previous studies on the influence of R-DPRs on stress granule assembly[24–26] and

the presence of UPF1 in stress granules[29], raising the possibility that NMD inhibition may be due to the sequestration of UPF1 and other NMD factors in stress granules.

To test whether DPRs can also induce stress granules in neurons, we expressed each DPR in mouse primary cortical neurons. Compared to those in cell lines, Upf1-positive stress granules were exceedingly rare in neurons (Supplementary Fig. 6). We detected Upf1-positive, Ataxin-2-positive stress granules only in a small fraction (<5%) of poly(GR)-expressing neurons (Supplementary Fig. 6), consistent with a recent study showing that poly(GR) co-localizes with stress granule-like inclusions in $(G_4C_2)_{149}$-expressing mouse brain[30]. These results suggest either that the neuronal cytoplasm may be less permissive to stress granule formation, or that neuronal stress granules may exhibit distinct morphology and/or dynamics from those in cell lines.

**Stress granules are not required for NMD inhibition.** To directly assess the role of stress granule formation in DPR-induced NMD inhibition, we expressed GFP, poly(GR), or poly (PR) in wild-type (WT) or G3BP1/2 double-knockout (G3BP-DKO) U2OS cells[31] (Supplementary Fig. 7a). Consistent with previous studies[24,25], while R-DPRs were expressed at similar levels between G3BP-WT and -DKO cells (Supplementary Fig. 7b), R-DPR-induced stress granule formation was substantially reduced in G3BP-DKO cells (Fig. 3c, Supplementary Fig. 7c). Despite the large reduction in stress granules in G3BP-DKO cells, R-DPRs still increased the abundance of NMD targets (Fig. 3d), suggesting that stress granule formation is not required for the inhibitory effect of R-DPRs on NMD.

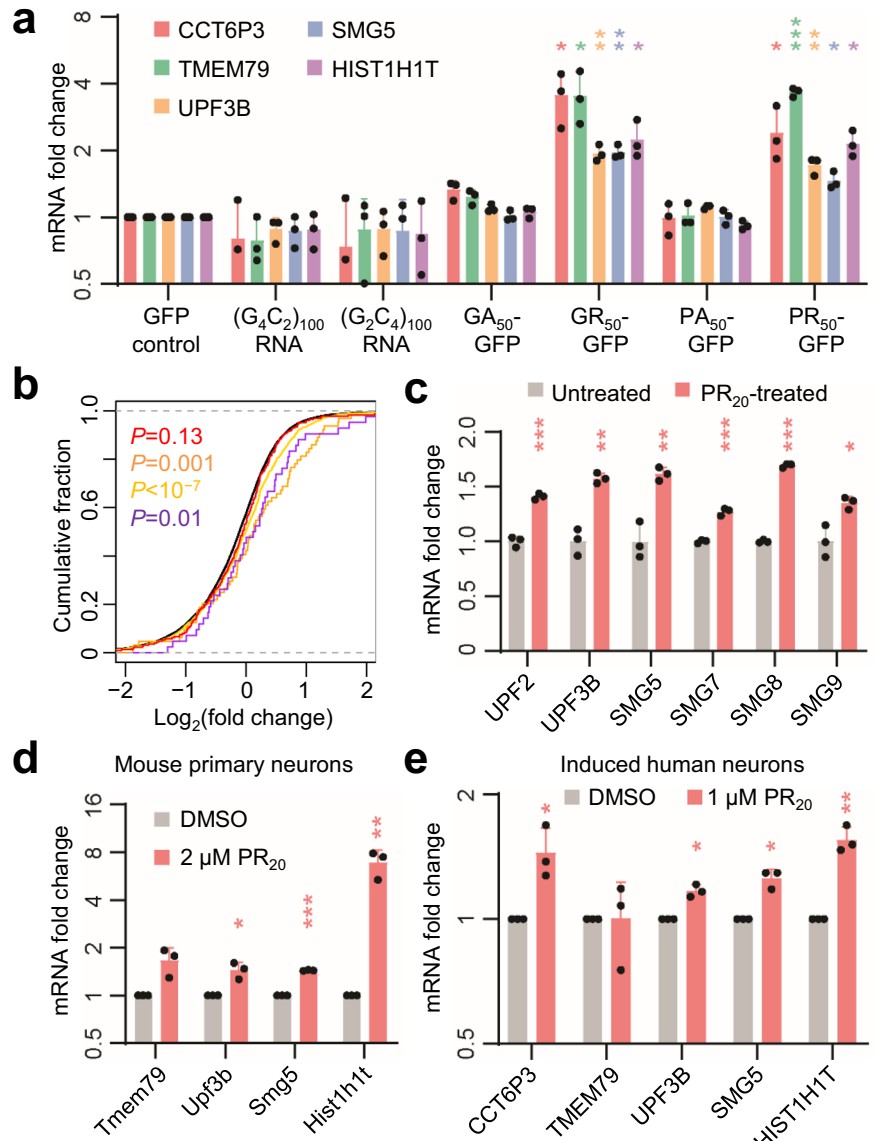

**Fig. 2 R-DPRs inhibit UPF1-dependent RNA decay in cultured cells. a** Changes in UPF1 target mRNA abundance in HEK293 cells ectopically expressing each repeat RNA or DPR. Expression levels relative to GAPDH were normalized to the GFP control. $n = 3$ independent experiments. **b** CDFs of changes in abundance for all mRNAs (black), NMD targets identified in Johnson et al. ($N = 275$, red), Tani et al. ($N = 76$, orange), in Colombo et al. ($N = 1271$, yellow), and canonical histone mRNAs (purple), comparing between PR20-treated and control K562 cells. $P$ values, two-sided Mann–Whitney $U$ tests. **c** Changes in NMD factor mRNA abundance between control ($n = 3$) and PR20-treated ($n = 3$) K562 cells. Data are presented as mean values ± SD. *$P < 0.05$; **$P < 0.01$; ***$P < 0.001$, two-sided unpaired $t$ tests. **d** Changes in abundance of UPF1 target mRNA orthologs in 2 μM PR20-treated mouse primary neurons. $n = 3$ independent experiments. **e** Change in UPF1 target abundance in 1 μM PR20-treated human iNeurons. $n = 3$ independent experiments. **a**, **d**, **e** Data are presented as mean values ± SD. *$P < 0.05$; **$P < 0.01$; ***$P < 0.001$, two-sided ratio $t$ tests. Source data and exact $P$ values are provided in the Source Data file.

In an orthogonal approach to reduce stress granule formation, we treated HEK293 cells with ISRIB (integrated stress response inhibitor), which has been shown to block the downstream effects of eIF2α phosphorylation including stress granule formation[32]. As expected, ISRIB treatment partially reduced stress granules in poly(PR)-expressing cells (Supplementary Fig. 8a). However, the increase in abundance of NMD targets was largely unchanged (Supplementary Fig. 8b). These results, together with those from G3BP-DKO cells, indicate that recruitment of UPF1 to stress granules is not the cause of NMD inhibition.

**R-DPRs inhibit NMD via translational repression.** $G_4C_2$ repeats and R-DPRs have also been associated with nucleocytoplasmic transport defects[33]. If nuclear export of RNA is delayed, both NMD targets and non-targets would be expected accumulate in the nucleus. We measured the relative abundance of UPF1 targets in the nuclear and cytoplasmic fractions, and found that both poly(GR) and poly(PR) caused UPF1 targets to accumulate predominantly in the cytoplasm (Fig. 4a). Therefore, nuclear RNA retention is not the cause of the accumulation of NMD targets either. In addition, R-DPR expression did not lead to significant changes in UPF1 phosphorylation levels (Supplementary Fig. 9).

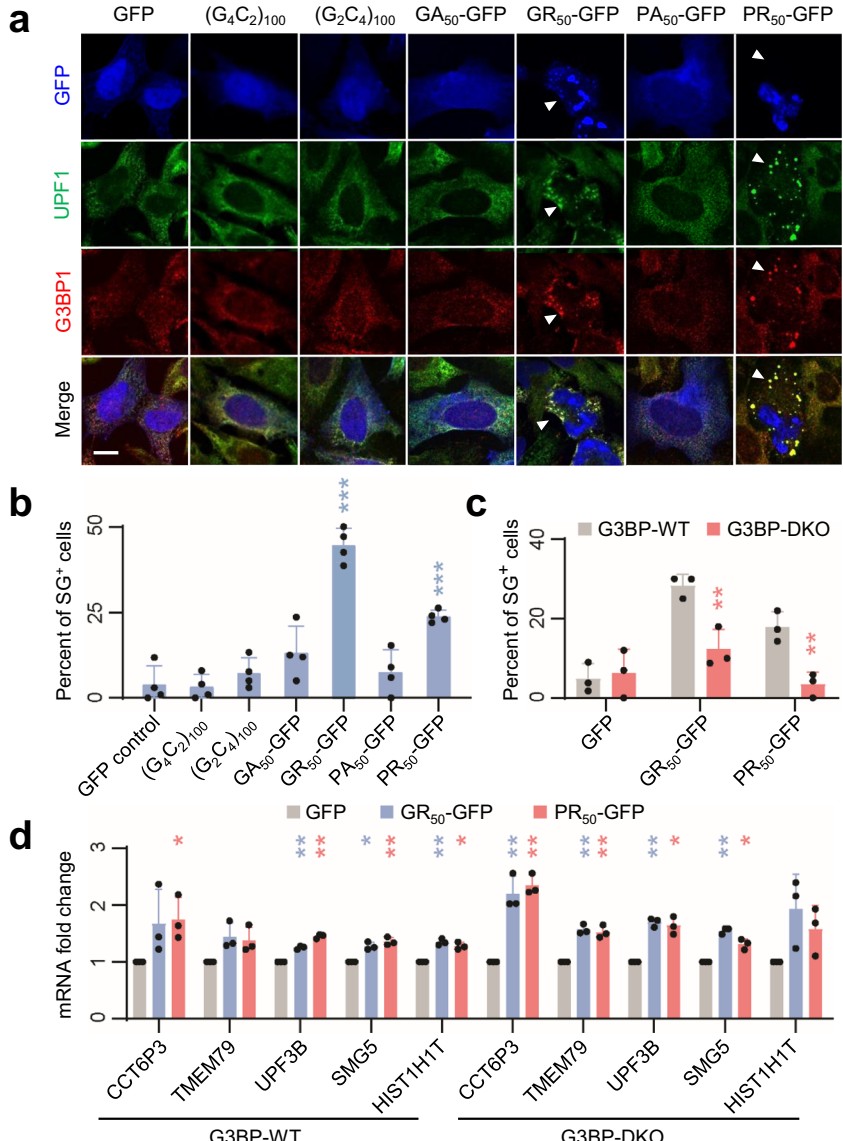

**Fig. 3 Recruitment of UPF1 to stress granules is independent of NMD inhibition. a** UPF1 and G3BP1 localization in HeLa cells expressing either GFP only, repeat RNA, or GFP-tagged DPR. Stress granules are indicated by arrowheads. Scale bar, 10 μm. Similar results were obtained from four independent experiments. **b** Quantification of stress granule (SG)-positive cells. $n = 4$ independent experiments. Data are presented as mean values ± SD. ***$P < 0.001$, two-sided unpaired $t$ tests. **c** Quantification of stress granule formation in G3BP-WT and -DKO cells. $n = 3$ independent experiments. Data are presented as mean values ± SD. **$P < 0.01$, two-sided unpaired $t$ tests. **d** Changes in UPF1 target mRNA abundance in G3BP-WT and -DKO cells. Expression levels were normalized to the GFP-only control. $n = 3$ independent experiments. Data are presented as mean values ± SD. *$P < 0.05$; **$P < 0.01$, two-sided ratio $t$ tests. Source data and exact $P$ values are provided in the Source Data file.

R-DPRs have previously been shown to inhibit translation both in vitro[34,35] and in vivo[36]. Because NMD is translation-dependent, global translational repression would be expected to inhibit NMD. We confirmed that poly(GR) and poly(PR) substantially inhibited global translation in HEK293 cells, as indicated by reduced puromycin incorporation into nascent peptides (Fig. 4b). To measure R-DPR-induced translation repression more quantitatively, we treated primary cortical neurons expressing firefly luciferase with several different concentrations of PR$_{20}$, and after 24 h, measured the ratio between luciferase activity and luciferase mRNA abundance. Indeed, significant translational repression was observed starting from the lowest PR$_{20}$ concentration (1 μM) (Fig. 4c).

Using this quantitative assay, we assessed the extent to which translational repression could explain the inhibitory effect of R-DPRs on NMD. We treated primary cortical neurons with either PR$_{20}$ or CHX at several different concentrations. After 24 h, we measured for each sample, the degree of translational repression and the increase in NMD target abundance (Fig. 4d). The relationship between the changes in NMD target abundance and the degree of translational repression was fitted by log–log models. For each of the three tested NMD targets, PR$_{20}$-induced NMD target accumulation was almost fully (92−122%) explained by translation repression (Fig. 4d). Therefore, the observed NMD inhibition is primarily a consequence of global translational repression by R-DPRs.

**UPF1 overexpression protects neurons from R-DPR toxicity.**
Consistent with the notion that R-DPRs cause neurodegeneration

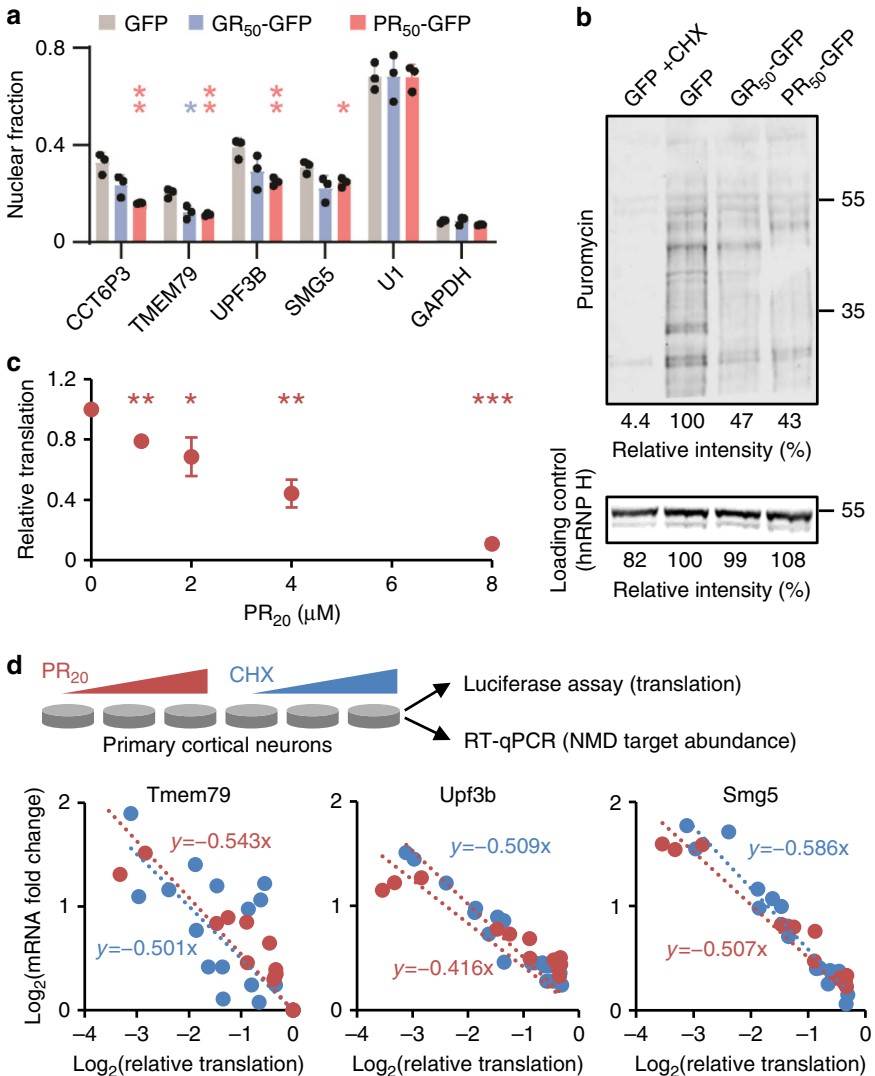

**Fig. 4 Translation repression is the primary cause of poly(PR)-induced NMD inhibition. a** Nuclear fractions of NMD targets, U1 snRNA, and GAPDH mRNA (cytoplasmic) in HEK293 cells expressing GFP, poly(GR), or poly(PR). $n = 3$ independent experiments. Data are presented as mean values ± SD. *$P < 0.05$; **$P < 0.01$, two-sided unpaired $t$ tests. **b** Puromycin-labeled nascent peptides in HEK293 cells expressing GFP, poly(GR), or poly(PR). +CHX, CHX added 1 h before puromycin labeling. Similar results were obtained from three independent experiments. **c** Global translation levels in primary neurons treated with PR$_{20}$ for 24 h, quantified by the activity of a transfected luciferase reporter, normalized to mRNA levels and untreated control. $n = 3$ independent experiments. Data are presented as mean values ± SD. *$P < 0.05$; **$P < 0.01$; ***$P < 0.001$, two-sided ratio $t$ tests. **d** Comparisons between the effects of PR$_{20}$ (red) and CHX (blue) on NMD targets. Degrees of translation repression and NMD target abundance were quantified in primary neurons treated with increasing concentrations of PR$_{20}$ or CHX. Zero-intercept linear functions were fit to normalized, log$_2$-transformed values. Source data and exact $P$ values are provided in the Source Data file.

in part through inhibiting NMD, recent studies have shown that overexpression of UPF1 and, to a lesser extent, UPF2 could reduce R-DPR neurotoxicity in flies[37,38]. To determine whether the UPF1 is also protective in mammalian neurons, we over-expressed UPF1 in primary cortical neurons before treating them with 2 μM PR$_{20}$. Consistent with previous studies[23], PR$_{20}$ caused cell death in >80% neurons within 48 h (Fig. 5). Compared to the control group, overexpression of wild-type UPF1 significantly increased neuronal survival by an average of 71% (Fig. 5).

UPF1 have multiple functions beyond NMD[20]. To determine whether the neuroprotective effect of UPF1 requires its NMD activity, we tested a variety of NMD-deficient UPF1 mutants[39], including C126S (deficient UPF2 binding), R854C (deficient helicase activity), G506R/G508E (GGRE, deficient ATPase/heli-case activity), and S1084A/S1089A/S1100A/S1107A (4SA, lacking four phosphorylation sites[40]). Using an NMD reporter assay, we

confirmed that only wild-type UPF1, but none of the four tested mutants could rescue NMD activity in UPF1-deficient cells (Supplementary Fig. 10). Correspondingly, none of the four NMD-deficient UPF1 mutants increased survival of PR$_{20}$-treated neurons (Fig. 5), suggesting that the neuroprotective effect of UPF1 requires its NMD function.

## Discussion
Emerging evidence suggests that widespread transcriptomic aberration is a distinguishing feature of c9ALS/FTD. Previous studies have largely attributed it to the deficiency of splicing factors (e.g., hnRNP H), leading to increased production rates of misprocessed RNAs[4,11]. Our results suggest that decreased degradation rates resulting from NMD inhibition also causes the accumulation of potentially deleterious RNAs in c9ALS/FTD.

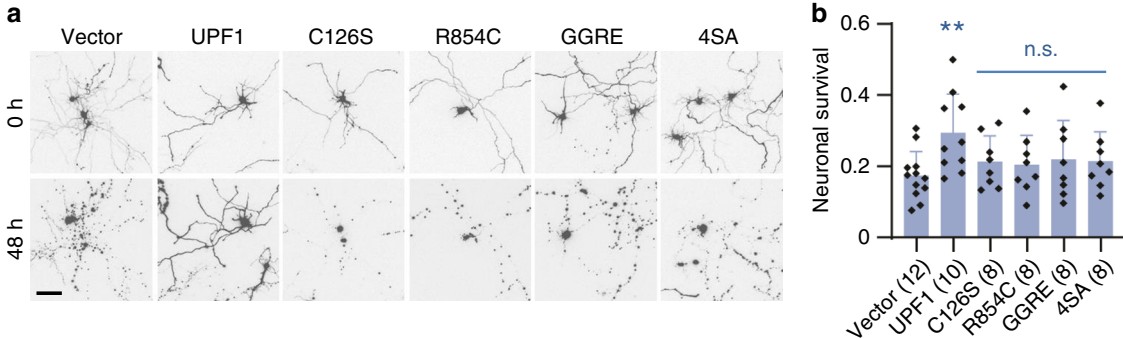

**Fig. 5 Overexpression of UPF1, but not NMD-deficient UPF1 mutants, reduces PR toxicity in primary neurons. a** Representative images of mApple-labeled primary cortical neurons expressing BFP (vector), UPF1, or NMD-deficient UPF1 mutants before and after treatment of 2 μM $PR_{20}$ for 48 h. Scale bar, 100 μm. **b** Quantification of neuronal survival after 48 h of $PR_{20}$ treatment. Numbers of independent experiments are indicated in parenthesis. Data are presented as mean values ± SD. **$P < 0.01$; n.s., not significant, two-sided paired $t$ tests. Source data and exact $P$ values are provided in the Source Data file.

While intron-specific splicing defects may also exist, the observed wide spectrum of increased intron retention in c9ALS subjects suggests that a more global defect in the degradation of mis-processed RNAs via NMD may underlie this phenomenon.

Aside from targeting misprocessed RNAs such as intron-retaining mRNAs (i.e., quality control), a separate function of NMD is lowering the abundance of many endogenous, properly processed RNAs (i.e., quantity control). We found in c9ALS subjects an overall upregulation of numerous NMD target mRNAs as well as many noncoding RNAs including pseudogene transcripts and the splice variants of small nucleolar RNA precursors[41]. Since many NMD factors including UPF1 are conserved in all eukaryotes and required for survival in most metazoans[42–44], chronic NMD inhibition would likely be detrimental to neuronal viability[13]. Indeed, loss-of-function mutations in both UPF2 and UPF3B have associated with intellectual disabilities[15,45], suggesting a heightened vulnerability of the nervous system to NMD deficiencies.

Our analysis showed that inhibition of RNA decay is not limited to NMD, but also affects other UPF1 targets such as canonical histone mRNAs[19,20]. The multiple functions of UPF1 beyond NMD may in part explain the recent observation that ectopic expression of UPF1 shows stronger neuroprotective effect in R-DPR fly models than that of UPF2, an NMD-specific factor[37]. Notably, aberrations in heterochromatin have been recently linked to poly(PR) expression in mouse models[46]. It will be interesting to determine whether the accumulated histone mRNAs are translated, and how they may impact histone turnover, chromatin modifications, and neuronal survival in c9ALS/FTD.

Ectopically expressed R-DPRs are sufficient to recapitulate NMD inhibition in cultured cells, suggesting that in c9ALS/FTD patients, NMD inhibition may be the consequence of low levels of poly(GR) and/or poly(PR) accumulation over a long period. Considering that R-DPRs are strong stress granule inducers and that UPF1 is a known stress granule component, it is tempting to believe that these DPRs inhibit NMD by sequestering UPF1 in stress granules and away from P-bodies and the rest of cytoplasm, where NMD may occur more efficiently. However, we obtained multiple lines of evidence arguing against this hypothesis: (i) In cell lines, poly(GR) promoted stress granule assembly more strongly than poly(PR), but these two R-DPRs inhibited NMD to similar degrees. (ii) R-DPR expression rarely caused stress granule assembly in neurons. (iii) Blocking stress granule assembly by deleting G3BP1/2 did not alleviate NMD inhibition. (iv) ISRIB treatment, which reduced poly(PR)-induced stress granules, had no effect on NMD target accumulation. These results collectively reject the hypothesis that stress granule assembly is a main contributor to NMD inhibition. Instead, stress granule formation and NMD inhibition appear to be two independent consequences of R-DPRs.

Previous studies have shown that R-DPRs cause global reduction in translation[36,47]. Considering that NMD requires translation to distinguish targets from non-targets, we reasoned that translation repression must at least in part contribute to R-DPR-induced NMD inhibition. From the dose-response relationships between NMD target abundance and translational repression caused by either $PR_{20}$ or CHX, we found that the effect of $PR_{20}$ on NMD is almost fully accounted for by its effect on translation. Therefore, at least for the several tested targets, translational repression appears to be the primary, if not exclusive, cause of NMD inhibition by R-DPRs. It remains to be tested whether this may be true at a transcriptome-wide level, and to what extent some NMD targets may be more sensitive to R-DPRs than others.

Although the mechanism of non-AUG translation producing DPRs in c9ALS/FTD remains unclear, poly(GA), part of poly (GP), and poly(GR) are potentially translated from the sense C9orf72 mRNA retaining its first intron, which itself may be an NMD target[38]. Therefore, NMD inhibition would in turn allow this intron-retaining mRNA to further accumulate, producing more DPRs. This vicious circle would presumably accelerate the accumulation of more deleterious RNAs and proteins in cells and ultimately cause cell death.

Recent studies has shown that ectopic expression of NMD factors can alleviate poly(GR)- and poly(PR)-induced toxicity in SH-SY5Y neuroblastoma cell line and in flies[38,48]. Our results extended these findings to mammalian neurons and provided new mechanistic insight into the neuroprotective effect of UPF1. Indeed, wild-type UPF1 significantly enhanced neuronal survival in the presence of synthetic $PR_{20}$ peptides. However, none of the NMD-deficient UPF1 mutants, including $UPF1^{C126S}$, which is deficient in UPF2 binding, could enhance neuronal survival. While additional contributions from the NMD-independent functions of UPF1 may exist, these results suggest a major role of NMD in the neuroprotective functions of UPF1. Considering that UPF1 and NMD have now been implicated in multiple ALS/FTD subtypes[48–50], restoring UPF1 and NMD activity to a physiological level may represent a new therapeutic approach.

## Methods

**RNA-seq data analysis**. FASTQ files of previously published RNA-seq datasets were downloaded from European Nucleotide Archive (ENA), and uniquely mapped to the human (GRCh38) or mouse (GRCm38) reference genome using STAR. Exon and intron read densities for all annotated genes were quantified using

BEDTools. After excluding genes with <1 read/sample on average, reads per million uniquely mapped reads (RPM) values were calculated for each gene. A pseudo-RPM value of 0.1 was added to all RPM values before calculating the fold change between ALS or experimental samples and controls. For NMD target analysis, three independent gene sets were used: (i) genes that are upregulated by ≥2 folds in Upf2-KO mouse forebrain with $P$ values <0.05 in Johnson et al. ($N = 275$); (ii) "group C" genes in Tani et al. ($N = 76$), (iii) genes with $P_{meta\_meta} < 10^{-5}$ in Colombo et al. ($N = 1271$).

**Primary neuronal culture.** Animals were cared for by the Yale Animal Resource Center. All experiments were approved by Yale's Institutional Animal Care and Use Committee (Protocol #2018–20207) and performed in accordance with the American Association for Accreditation of Laboratory Animal Care (AAALAC). Brain cortices were dissected from embryonic day 15 (E15) pups from CO$_2$-euthanized pregnant C57/Bl6 mice (Charles River Laboratories), dissociated by incubating in 0.25% trypsin for 10 min at 37 °C, followed by gentle trituration in DMEM supplemented with 10% FBS (ThermoFisher). Neurons were pelleted and resuspended in Neurobasal medium supplemented with 2% B27 (ThermoFisher), 1% GlutaMAX, 33 mM glucose, and 37.5 mM NaCl, and plated at ~300,000 cells/well in glass-bottomed 24-well plates (Cellvis) coated with poly-L-lysine (Sigma) and mouse laminin (ThermoFisher).

**RT-qPCR.** Expression constructs for repeat RNAs or GFP-tagged codon-optimized DPRs were transfected in HEK293 cells or U-2 OS cells with Lipofectamine 2000 (Invitrogen) or FuGENE HD (Promega), respectively, according to the manufacturers' instructions. 24−48 h after transfection, total RNA was extracted with TRIzol and treated with Turbo DNase (Invitrogen). RT-qPCR was performed on a CFX96 RT-PCR system (Bio-Rad) with Luna Universal One-Step RT-qPCR reagents (New England Biolabs). Fifty nanograms of total RNA and 250 nM primers (see Source Data file for primer sequences) were added in each reaction. Ct values were averaged across two technical replicates and normalized to GAPDH internal controls. Three independent replicate experiments were typically performed.

**Western blotting.** Cultured cells were lysed in RIPA buffer on ice for 10 min. After 10-min centrifugation at 4 °C, 20,000 × g, whole cell lysates were mixed with 4X LDS sample buffer (Invitrogen) and denatured at 95 °C for 5 min. Samples were loaded on a 4–12% Bis-Tris SDS-PAGE gel, run at 200 V for 45 min in MOPS buffer, and transferred onto a nitrocellulose membrane (Bio-Rad) in an XCell II Blot module (Invitrogen) (15 V, 45 min). After 1-h blocking with 5% nonfat dry milk in PBST, the membrane was incubated with primary antibodies (rabbit anti-UPF1, Cell Signaling Technology #12040; mouse anti-G3BP1, Millipore #05–1938; mouse anti-puromycin clone 12D10, Millipore #MABE343; rabbit anti-hnRNP H, Bethyl Laboratories #A300-511A; chicken anti-GFP, Aves Labs #1010; rabbit anti-phospho-UPF1, Millipore #07-1016; mouse anti-β-Actin, Cell Signaling Technology #4967; rabbit anti-G3BP2, Bethyl Laboratories #A302-040A; rabbit anti-BFP, Evrogen #AB233) diluted (1:2000) in 5% milk/PBST at 4 °C with slow shaking overnight. After incubation, membranes were rinsed three times with PBST, and incubated with IR680- or IR800-conjugated secondary antibodies (Li-Cor) diluted (1:10,000) in 5% milk/PBST at room temperature for 1 h. After three rinses with PBST, membranes were imaged using an Odyssey CLx system (Li-Cor).

**Immunocytochemistry.** Transfected HEK293 cells, HeLa cells, U2OS cells, or E15 mouse cortical neurons on coverslips were fixed at room temperature with 4% paraformaldehyde in PBS, washed three times with PBS, and permeabilized with 0.5% Triton X-100. After washing and 1-h blocking with 10% goat serum, cells were incubated with primary antibodies (rabbit anti-UPF1, Cell Signaling Technology #12040; mouse anti-G3BP1, Millipore #05-1938; rabbit anti-UPF3B, Invitrogen # PA5-51652; rabbit anti-TIAR, BD Biosciences #610352; rabbit anti-Ataxin-2, BD Biosciences #611378) diluted (1:1000) in 5% BSA at 4 °C overnight. After washing, cells were incubated for 1 h at room temperature with Alexa Fluor-conjugated secondary antibodies (Invitrogen) diluted (1:10,000) in 5% BSA. After washing, cells were stained with Hoechst 33342, mounted with ProLong Diamond Antifade Reagent, and imaged on a Nikon Ti-E Eclipse inverted microscope (spinning disc confocal) with a ×60 oil objective (Olympus).

**Quantitation of translational repression.** E15 mouse primary cortical neurons were transiently transfected with a plasmid expressing firefly luciferase. After 6-8 h, DMSO, CHX (0.0075~0.24 μg/mL), or PR$_{20}$ (0.5~8 μM) was added to media. Cells were lysed after 24 h, with half of the lysate used for luciferase assays (Promega) and the half for RNA extraction. Luciferase mRNA and NMD target abundance were measured by qRT-PCR. Translational activity was first calculated as the ratio between luciferase activity and luciferase mRNA abundance, then normalized to the DMSO control sample.

**Neuronal viability analysis.** Primary cortical neurons were transfected using Lipofectamine 2000 (Invitrogen) with a 1:9 ratio of an mApple-expressing plasmid (Addgene #54567) and an expression construct for BFP only, wild-type or mutant UPF1-BFP. 48 h after transfection, PR$_{20}$ peptides were added to the media at a final

concentration of 2 μM. Neurons were imaged 0 or 48 h after PR$_{20}$ treatment on a BioTek Lionheart FX automated microscope with a ×4 objective and quantified using the Gen5 image analysis software.

**Reporting summary.** Further information on research design is available in the Nature Research Reporting Summary linked to this article.

## Data availability

The data that support this study are available from the corresponding author upon reasonable request. RNA-seq data are available through Gene Expression Omnibus (GEO) under accession codes GSE67196 and GSE109177. Microarray data are available through ArrayExpress under accession code E-MTAB-1926. The source data underlying Figs. 2–5 and Supplementary Figs. 3, 4, 7–10 are provided as a Source Data file.

## Code availability

In-house shell scripts and R codes used for analyzing RNA-seq data are available upon request.

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

## Acknowledgements

We thank D. Trotti for DPR expression constructs, N. Kedersha and P. Anderson for G3BP-WT and -DKO cell lines, C. Reddy and P. Gopal for assistance in primary neuronal culture, W. Hancock-Cerutti and P. De Camilli for iNeuron culture, J. Steitz, A. Alexandrov, A. Horwich, S. Strittmatter, J. Lim, S. Chandra, J. Hogg, and members of the Guo lab for discussions and comments on the manuscript. This work was supported by an NIH New Innovator Award (DP2 GM132930), the Muscular Dystrophy Association (MDA602934), the Ludwig Family Foundation, and the Yale Scholar in Neuroscience Fund. J.U.G. is a NARSAD Young Investigator and a Klingenstein−Simons Fellow in Neuroscience.

## Author contributions

Y.S. and J.U.G. designed the study. Y.S. led the project, performed, and analyzed all the experiments. A.E. assisted in the immunocytochemistry analysis. J.Z., A.U.I., and J.U.G. performed the computational analyses of RNA-seq data. J.U.G. wrote the manuscript with input from all authors.

## Competing interests

The authors declare no competing interests.
