## [Peer Review File · Nature Communications]

Reviewers' comments:

Reviewer #1 (Remarks to the Author):

The authors re-analyzed published sequencing data from postmortem brain tissue of C9ALS and sALS patients and compared to unaffected individuals. They compared these data against previously published RNA-sequencing data that identify mRNA transcripts that are increased in abundance in response to knockout of UPF1 and/or other NMD machinery in HeLa cell lines, observing that these “endogenous NMD targets” are generally increased in abundance in the brains of C9ALS patients. The authors claim that this observation hints at a potential defect in UPF1-mediated mRNA decay. Indeed, a handful of these NMD targets increase in abundance upon introduction of arginine-rich DPRs (via transfection or addition of synthetic peptides) in U2OS, HeLa, and mouse cortical neurons. In pursuit of mechanism, the authors conclude that UPF1 localization to poly(GR) or poly(PR) induced stress granules is independent of this NMD dysfunction after a short series of imaging and qRT-PCR experiments. The presented experiments are well designed, although the results are overstated and sometimes unfounded.

Overall, this study represents potentially interesting preliminary findings that, when followed up with additional mechanistic experiments in more relevant cell types, has the potential to provide an interesting data interpretation on previously published reports. However, there are concerns regarding the validity, relevance, and interpretation of the presented data.

Major concerns:

1. The authors assert that NMD substrates are accumulated in c9ALS brains. However, the samples used for this original sequencing were not exclusively neuronal. Therefore, the observation that NMD substrates are accumulated in these brains does not mean they are accumulated in affected neurons. Moreover, the lists of NMD targets used for comparison are derived from HeLa cells. There is no existing evidence in the literature that NMD targets are conserved between these immortalized cell lines and neurons. In fact, there exists evidence that NMD function does not occur to the same extent in different cell types (Linde et al., 2007 PubMed ID: 17625509). Therefore, it is difficult to believe that the authors' assertion of a global increase in mRNA target abundance in c9ALS brains is valid.
2. In several experiments, exogenous constructs are used to overexpress the products of the c9orf72 repeat expansion. Since one of the mentioned constructs (poly(GP)) could not be validated, this raises concerns about whether these other constructs are valid. A comparison of the levels of expression of each construct via Western blot (not imaging-based validation) is necessary to determine whether the observed differences between various DPRs is due potentially to differences in expression level. Additionally, the use of overexpression models without validation in a truly endogenous model of c9ALS (i.e. iPSC-derived motor neurons or postmortem tissue—which are widely available) raises significant concerns as to the relevance of the authors' conclusions.
3. Recent studies have highlighted functions of UPF1 distinct from its known roles in NMD (Ryu et al., 2019 PubMed ID: 30584064). Thus, overexpression of UPF1, as performed for Figure 2E in mouse neurons, may be affecting neuronal health independent of NMD. Parsing out a way to specifically activate the NMD function of UPF1 will be important to truly know if the observed rescue is due to

restored NMD function, not some other pathway in which UPF1 is involved.

4. The assertion that the mechanism for DPR-mediated dysfunction of NMD involves translation inhibition is unsupported by the presented data. It is known that translation elongation is necessary for NMD to occur, as the ribosome must reach a termination codon for UPF1 to be phosphorylated and signal downstream degradation, so the results shown in Figure 4B are expected. However, the experiment performed to show that translation inhibition is upstream of NMD dysfunction is incorrect. 50ug/mL of CHX is a very high concentration, and will likely stop all translation. Thus, it is obvious that no further perturbation to translation would have an additional effect on NMD target abundance. The proper experiment required to show that translation inhibition is the perpetrator of NMD dysfunction is to increase global translation in the presence of the overexpressed DPRs to observe whether the mRNA abundance NMD targets return to control levels.

Minor concerns:

1. Since UPF1 must be phosphorylated for downstream degradation of an NMD target, it is important to know whether UPF1 phosphorylation is altered under any of the conditions presented within this manuscript. Phospho-UPF1 antibodies are publicly available. An increase in total UPF1 expression paired with a decrease in pUPF1 presence would help support the authors' claims that NMD is inhibited.
2. Images to pair with Figure 3C would be useful, similar to what is shown in 3A.
3. The level of DPR overexpression in G3BP-WT versus G3BP-DKO cells looks vastly different by imaging (specifically poly(GR)). Again, a Western blot confirming equivalent levels of overexpression would be helpful to correctly interpret results. Additionally, many of the GFP panels in Supp. Fig. 4 look very overexposed.
4. No statistical analysis is shown for Fig. 3D, 4A, or 4C. This analysis is important to support/reject the authors' claim that UPF1 targets accumulate in the cytoplasm.
5. Since the authors suggest that the repeat RNA is potentially an NMD substrate, they should perform the experiment(s) to support/reject the claim.
6. 0.5% Triton X-100 is relatively high for permeabilization of cells and may affect the observation of stress granules via imaging. It is recommended that a lower concentration of detergent (0.1-0.2%) be used for any imaging of stress granules.
7. The authors use G3BP1/2 double knockout cell lines, but do not validate that either protein is absent. Validation of these cell lines by Western Blot is necessary to interpret their findings.
8. Use of a G3BP-DKO cell line, though not neuronal, is a clever way to address the importance of stress granule formation in the observed inhibition of NMD by DPR overexpression. However, as the authors note, G3BP-negative (TIAR-positive) stress granules still form in the absence of G3BP. Thus, use of an orthogonal method to prevent stress granule formation (ISRIB treatment, for example) will be important to validate that loss of stress granule formation does not alter the authors' observed NMD inhibition.

Reviewer #2 (Remarks to the Author):

Sun and colleagues report in this manuscript the inhibition of UPF1-mediated RNA decay by dipeptide

repeats in an independent manner of stress granule. To address this, the authors observed UPF1-mediated RNA decay targets in c9ALS brains and the expression of DPRs inhibited NMD. Finally, UPF1 was enriched in the stress granule by DPRs, which was not caused by NMD inhibition. Overall, the authors suggested that DPRs increased the enrichment of UPF1 in the stress granule, resulting in NMD inhibition. Although this is a potentially interesting manuscript, there are some technical issues and the provided evidences does not support the authors' conclusions. Furthermore, the results in this manuscript are preliminary. To be suitable in Nature communication, the authors must provide i) how UPF1 is sequester in the stress granule by DPRs, ii) why UPF1 in the stress granule is not functional, and iii) the responses the below concerns.

1. Figure 1: The authors seemed to define NMD targets obtained from the previous reports, where HeLa cells were employed. However, each type of cells has different transcriptome by alternative splicing, the authors must define PTC-triggering NMD targets by mapping sequences in c9ALS or neuronal cells.
2. Figure 2: The authors insisted that NMD inhibition increased DPR toxicity by because overexpression of UPF1 rescued cell growth inhibited by DPRs. However, it is impossible the mere UPF1 overexpression increased NMD efficiency. To confirm whether UPF1 is involved in preventing DPR toxicity, the authors need to employ other UPF1 dominant negative mutants including unphosphorylated or null helicase activity UPF1 and/or UPF1 deletion mutants.
3. Figure 3: The authors indicated that DPRs increased the sequestration of UPF1 in the stress granule, which is the reason why NMD is inhibited. However, UPF1 was still observed in the cytoplasmic region (not stress granule) even by DPRs expression (Fig. 3a) and I am wondering whether those UPF1 is sufficient or lack for NMD. To address this, the authors need to develop and use UPF1 mutants that cannot go into the stress granule by DPRs but are functional in NMD. I do not understand the sentence "recruitment of UPF1 to stress granule is not eh cause of NMD inhibition" because knockout of G3BP did not represent NMD inhibition.
4. Figure 4: The authors need to provide the cell fractionation was complete.

We thank both reviewers for their positive feedback and constructive critiques on our manuscript. Following their suggestions, we have performed additional experimental and computational analyses to address all the questions. The results from these analyses further strengthened our conclusions, for example:

1. In addition to the previous lists of NMD targets defined in non-neuronal cell lines, we used a more relevant NMD target list (up-regulated transcripts in the *Upf2*-knockout mouse forebrain). Consistent with our previous findings using NMD target lists from HeLa cells, we observed accumulation of these putative neuronal NMD targets in c9ALS patients and iPSC-derived motor neurons.
2. In an orthogonal approach to reduce stress granule formation, we treated cells with ISRIB. Consistent with our previous results from G3BP1/2-knockout cells, ISRIB treatment also did not affect NMD inhibition by poly(PR), further arguing against a role for stress granules in causing the observed NMD deficits.
3. We quantified the extent to which NMD inhibition by poly(PR) could be explained by translational repression. By titrating the concentrations of PR₂₀ and CHX, we found that the increase in target abundance was almost fully explained by translation repression.
4. We tested a variety of NMD-deficient UPF1 mutants for their ability to enhance neuronal survival using the PR₂₀ toxicity assay. None of these mutants exhibited the neuroprotective effect of wild-type UPF1, suggesting that the neuroprotective function of UPF1 requires its NMD function.

Below are our point-by-point responses to the reviewers' comments.

Reviewer #1

The authors re-analyzed published sequencing data from postmortem brain tissue of C9ALS and sALS patients and compared to unaffected individuals. They compared these data against previously published RNA-sequencing data that identify mRNA transcripts that are increased in abundance in response to knockout of UPF1 and/or other NMD machinery in HeLa cell lines, observing that these "endogenous NMD targets" are generally increased in abundance in the brains of C9ALS patients. The authors claim that this observation hints at a potential defect in UPF1-mediated mRNA decay. Indeed, a handful of these NMD targets increase in abundance upon introduction of arginine-rich DPRs (via transfection or addition of synthetic peptides) in U2OS, HeLa, and mouse cortical neurons. In pursuit of mechanism, the authors conclude that UPF1 localization to poly(GR) or poly(PR) induced stress granules is independent of this NMD dysfunction after a short series of imaging and qRT-PCR experiments. The presented experiments are well designed, although the results are overstated and sometimes unfounded.

Overall, this study represents potentially interesting preliminary findings that, when followed up with additional mechanistic experiments in more relevant cell types, has the potential to provide

an interesting data interpretation on previously published reports. However, there are concerns regarding the validity, relevance, and interpretation of the presented data.

We thank this reviewer for considering our experiments well designed, and for his/her helpful suggestions on new analyses and data interpretation, which substantially improved our manuscript.

Major concerns:

1. *The authors assert that NMD substrates are accumulated in c9ALS brains. However, the samples used for this original sequencing were not exclusively neuronal. Therefore, the observation that NMD substrates are accumulated in these brains does not mean they are accumulated in affected neurons. Moreover, the lists of NMD targets used for comparison are derived from HeLa cells. There is no existing evidence in the literature that NMD targets are conserved between these immortalized cell lines and neurons. In fact, there exists evidence that NMD function does not occur to the same extent in different cell types (Linde et al., 2007 PubMed ID: 17625509). Therefore, it is difficult to believe that the authors' assertion of a global increase in mRNA target abundance in c9ALS brains is valid.*

As the reviewer correctly pointed out, the RNA-seq data we analyzed were obtained from frontal cortex tissues, which contain a variety of neuronal and glial cell types. In the original manuscript, we showed that synthetic PR₂₀ peptide treatment of cultured mouse cortical neurons increased NMD target abundance (**Fig. 3d**). To validate this finding in human neurons, we treated human iPSC-derived iNeurons with PR₂₀, and observed similar accumulation of putative NMD targets (new **Fig. 3e**). Furthermore, the accumulation of NMD targets were recapitulated in human iPSC-derived motor neurons (new **Supplementary Fig. 2**). Finally, to confirm that these selected transcripts are indeed NMD targets, we treated iNeurons with two mechanistically distinct NMD inhibitors, caffeine (UPF1 phosphorylation inhibitor) and CHX (translation inhibitor). Both NMD inhibitors increased the abundance of these putative NMD targets (new **Supplementary Fig. 4**), suggesting that they are indeed NMD targets in human neurons. Having said that, we did not intend to rule out the possibility that NMD inhibition may also occur in glia, which contribute significantly to c9ALS pathophysiology.

Indeed, NMD targets and/or efficiency may be different between cell types. However, these differences would only diminish, rather than increase, the signal when we compared NMD targets defined in other cell types between c9ALS, sALS, and control tissues. Following the reviewer's suggestion, we tested a third list of NMD targets identified in a more relevant tissue, which are the orthologs of upregulated mRNAs in *Upf2* KO mouse forebrain (Johnson et al., 2019). These mRNAs presumably contain both direct and indirect targets of UPF2-dependent NMD in the mouse cortex. As expected, these neuronal NMD target mRNAs also showed specific accumulation in c9ALS but not sALS brain samples (revised **Fig. 1b**). Consistent changes in NMD target abundance were observed in C9-iPSC-derived motor neurons but not

SOD1^{D90A} motor neurons (new **Supplementary Fig. 2**). Together, these results further strengthened our finding of NMD deficits in c9ALS.

2. *In several experiments, exogenous constructs are used to overexpress the products of the c9orf72 repeat expansion. Since one of the mentioned constructs (poly(GP)) could not be validated, this raises concerns about whether these other constructs are valid. A comparison of the levels of expression of each construct via Western blot (not imaging-based validation) is necessary to determine whether the observed differences between various DPRs is due potentially to differences in expression level. Additionally, the use of overexpression models without validation in a truly endogenous model of c9ALS (i.e. iPSC-derived motor neurons or postmortem tissue—which are widely available) raises significant concerns as to the relevance of the authors' conclusions.*

We apologize for the ambiguous statement regarding the validity of DPR constructs. We chose not to include the poly(GP) construct that could not be validated by Sanger sequencing, presumably due to the exceedingly high tendency of the coding sequence (GGNCCN)₅₀ to form secondary structures. However, the other four DPR constructs had no such issue and were fully validated by Sanger sequencing. Following this reviewer's suggestion, we included the western blot showing the expression levels of DPR-GFP constructs (new **Supplementary Fig. 3**). As expected from their inhibitory effects on translation, GR and PR were expressed at lower levels than GA and PA, thereby ruling out the possibility that the differential DPR effects on NMD or on stress granule formation may be due to their differential expression levels.

We appreciate the suggestion that we validate our findings in endogenous c9ALS models such as iPSC-derived motor neurons or postmortem tissues. Our initial observations of NMD target accumulation were made indeed using data from c9ALS postmortem tissues (**Fig. 1**). Following the reviewer's suggestion, we analyzed NMD target and histone mRNA abundance in iPSC-derived motor neurons from normal, c9ALS, and SOD1^{D90A}-ALS subjects (Donnelly et al., 2013 PMID: 24139042). Consistent with our results from post-mortem tissues, we observed significant NMD target and histone mRNA accumulation in c9ALS-iPSC-derived motor neurons but not SOD1^{D90A} motor neurons (new **Supplementary Fig. 2**), further strengthening our conclusion.

3. *Recent studies have highlighted functions of UPF1 distinct from its known roles in NMD (Ryu et al., 2019 PubMed ID: 30584064). Thus, overexpression of UPF1, as performed for Figure 2E in mouse neurons, may be affecting neuronal health independent of NMD. Parsing out a way to specifically activate the NMD function of UPF1 will be important to truly know if the observed rescue is due to restored NMD function, not some other pathway in which UPF1 is involved.*

The reviewer correctly pointed out that UPF1 has additional functions beyond NMD. Indeed, we showed in the manuscript that NMD-independent functions of UPF1, such as histone mRNA decay, may also be affected in c9ALS. Regarding the neuroprotective function of UPF1 against PR₂₀ toxicity, we now show in the revised manuscript that none of the tested NMD-deficient UPF1 mutants, including C126S (deficient UPF2 binding), R854A (deficient helicase activity), G506R/G508E (deficient ATPase/helicase activity), and S1084A/S1089A/S1100A/S1107A (4SA, lacking four phosphorylation sites), could rescue neuronal survival (new **Fig. 5**). While we cannot exclude potential contributions from NMD-independent functions of UPF1, these findings, along with a recent study showing that both UPF1 and UPF2 overexpression could reduce GR/PR toxicity in flies (Xu et al., 2019), strongly suggest that NMD functions of UPF1 are tightly coupled with its neuroprotective functions. We added the discussion of the potential contributions from NMD-independent functions of UPF1 in the revised manuscript.

4. The assertion that the mechanism for DPR-mediated dysfunction of NMD involves translation inhibition is unsupported by the presented data. It is known that translation elongation is necessary for NMD to occur, as the ribosome must reach a termination codon for UPF1 to be phosphorylated and signal downstream degradation, so the results shown in Figure 4B are expected. However, the experiment performed to show that translation inhibition is upstream of NMD dysfunction is incorrect. 50ug/mL of CHX is a very high concentration, and will likely stop all translation. Thus, it is obvious that no further perturbation to translation would have an additional effect on NMD target abundance. The proper experiment required to show that translation inhibition is the perpetrator of NMD dysfunction is to increase global translation in the presence of the overexpressed DPRs to observe whether the mRNA abundance NMD targets return to control levels.

We agree with the reviewer that the effect of high concentration of CHX on NMD targets (original Fig. 4b) are as expected. Considering that *i*) translational inhibition is sufficient to inhibit NMD (**Supplementary Fig. 4**), and *ii*) GR/PR inhibit translation (new **Fig. 4b & 4c**), we reasoned that GR/PR must inhibit NMD at least in part through translational repression. In the revised manuscript, we provide a more quantitative analysis to assess the extent to which translation repression may explain NMD inhibition by poly(PR) (new **Fig. 4d**). We treated cortical neurons with a series of concentrations of either PR₂₀ or CHX, and measured the degrees of translation inhibition and NMD target abundance at each concentration. We then compared the response curves of each NMD target between PR₂₀ and CHX treatments, normalized by the level of translation repression. For the tested NMD targets, we found that the PR₂₀-induced increase in mRNA abundance can be almost completely accounted for by the equivalent CHX treatment, namely, CHX treatment that inhibits translation to a similar degree. While some transcript-specific effects may exist and can be uncovered by future RNA-seq experiments, our results suggest that for the majority of NMD targets, translation inhibition is the primary cause of poly(PR)-induced NMD deficits.

Minor concerns:

1. Since UPF1 must be phosphorylated for downstream degradation of an NMD target, it is important to know whether UPF1 phosphorylation is altered under any of the conditions presented within this manuscript. Phospho-UPF1 antibodies are publicly available. An increase in total UPF1 expression paired with a decrease in pUPF1 presence would help support the authors' claims that NMD is inhibited.

Following the review's suggestion, we quantified UPF1 phosphorylation levels in HEK293 cells expressing poly(GR) or poly(PR) (new **Supplementary Fig. 9**). We did not observe significant changes in UPF1 phosphorylation, suggesting that NMD inhibition most likely occurred at the level of target selection, rather than UPF1 phosphorylation or dephosphorylation.

2. Images to pair with Figure 3C would be useful, similar to what is shown in 3A.

The images that correspond to **Fig. 3c** are presented in **Supplementary Fig. 7c**.

3. The level of DPR overexpression in G3BP-WT versus G3BP-DKO cells looks vastly different by imaging (specifically poly(GR)). Again, a Western blot confirming equivalent levels of overexpression would be helpful to correctly interpret results. Additionally, many of the GFP panels in Supp. Fig. 4 look very overexposed.

We have now replaced the original images with non-saturated ones (**Supplementary Fig. 7c**). Also following the reviewer's suggestion, we added the Western blots showing that GFP, poly(GR), and poly(PR) were expressed at similar levels in G3BP-WT and -DKO cells (**Supplementary Fig. 7b**).

4. No statistical analysis is shown for Fig. 3D, 4A, or 4C. This analysis is important to support/reject the authors' claim that UPF1 targets accumulate in the cytoplasm.

We have now added the statistical results to these figures.

5. Since the authors suggest that the repeat RNA is potentially an NMD substrate, they should perform the experiment(s) to support/reject the claim.

Following the review's suggestion, we tested the effect of NMD inhibitors on C9orf72 intron RNA abundance in three independent lymphoblastoid cell lines from c9ALS patients. Although

treating these cells with caffeine for 24 hours caused a substantial increase, CHX treatment resulted in a much smaller (17%) and statistically nonsignificant increase (**Fig. R1**). Therefore, we do not have conclusive evidence to either support or reject the notion that NMD is a major determinant of the abundance of C9orf72 intron-retaining RNA. While this notion is tangential to our current study, it is certainly worth future investigation.

6. 0.5% Triton X-100 is relatively high for permeabilization of cells and may affect the observation of stress granules via imaging. It is recommended that a lower concentration of detergent (0.1-0.2%) be used for any imaging of stress granules.

In the ISRIB treatment analysis (further discussed below), we tested the effect of Triton X-100 concentration on stress granule quantification. We found that reducing the concentration of Triton X-100 did not substantially change our results (**Fig. R2**). Therefore, we combined the results obtained at different Triton X-100 concentrations in **Supplementary Fig. 8a**.

7. The authors use G3BP1/2 double knockout cell lines, but do not validate that either protein is absent. Validation of these cell lines by Western Blot is necessary to interpret their findings.

Following the reviewer's suggestion, we have now added the Western blots confirming the lack of G3BP1/2 expression in G3BP-DKO cells (new **Supplementary Fig. 7a**).

8. Use of a G3BP-DKO cell line, though not neuronal, is a clever way to address the importance of stress granule formation in the observed inhibition of NMD by DPR overexpression. However, as the authors note, G3BP-negative (TIAR-positive) stress granules still form in the absence of G3BP. Thus, use of an orthogonal method to prevent stress granule formation (ISRIB treatment, for example) will be important to validate that loss of stress granule formation does not alter the authors' observed NMD inhibition.

We thank the reviewer for this great suggestion. The results are now provided in the new **Supplementary Fig. 8**. We found that ISRIB treatment indeed significantly reduced stress granules caused by poly(PR) (**Supplementary Fig. 8a**). However, NMD target accumulation was largely unchanged (**Supplementary Fig. 8b**). Therefore, these results are fully consistent with the G3BP-DKO results, and further strengthen our conclusion that DPR-induced NMD inhibition is stress granule-independent.

Reviewer #2:

Sun and colleagues report in this manuscript the inhibition of UPF1-mediated RNA decay by dipeptide repeats in an independent manner of stress granule. To address this, the authors observed UPF1-mediated RNA decay targets in c9ALS brains and the expression of DPRs inhibited NMD. Finally, UPF1 was enriched in the stress granule by DPRs, which was not caused by NMD inhibition. Overall, the authors suggested that DPRs increased the enrichment of UPF1 in the stress granule, resulting in NMD inhibition. Although this is a potentially interesting manuscript, there are some technical issues and the provided evidences does not support the authors' conclusions. Furthermore, the results in this manuscript are preliminary. To be suitable in Nature communication, the authors must provide i) how UPF1 is sequester in the stress granule by DPRs, ii) why UPF1 in the stress granule is not functional, and iii) the responses the below concerns.

Although it is currently unknown how UPF1 is recruited to stress granules, our conclusion is that its recruitment to stress granules is NOT the cause of NMD inhibition, as further discussed below. Therefore, we believe that the mechanism of UPF1 recruitment to stress granules and the functionality of UPF1 inside stress granules are both tangential to our current study.

1. Figure 1: The authors seemed to define NMD targets obtained from the previous reports, where HeLa cells were employed. However, each type of cells has different transcriptome by alternative splicing, the authors must define PTC-triggering NMD targets by mapping sequences in c9ALS or neuronal cells.

We agree with the reviewer that NMD targets and/or efficiency may be different between cell types. However, these differences would only diminish, rather than increase, the signal when we compared NMD targets defined in other cell types between c9ALS, sALS, and control tissues. Following the reviewer's suggestion, we tested a third list of NMD targets identified in a more relevant tissue, which are the orthologs of upregulated mRNAs in *Upf2* KO mouse forebrain. These mRNAs presumably contain both direct and indirect targets of UPF2-dependent NMD in the mouse cortex. As expected, these neuronal NMD target mRNAs also showed specific accumulation in c9ALS but not sALS brain samples (revised **Fig. 1b**). Furthermore, similar changes in NMD target abundance were observed in C9-iPSC-derived motor neurons but not SOD1^{D90A} motor neurons (new **Supplementary Fig. 2**). Together, these results further strengthened our finding of NMD deficits in c9ALS.

2. Figure 2: The authors insisted that NMD inhibition increased DPR toxicity by because overexpression of UPF1 rescued cell growth inhibited by DPRs. However, it is impossible the mere UPF1 overexpression increased NMD efficiency. To confirm whether UPF1 is involved in preventing DPR toxicity, the authors need to employ other UPF1 dominant negative mutants including unphosphorylated or null helicase activity UPF1 and/or UPF1 deletion mutants.

Following the review's suggestion, we tested a variety of NMD-deficient UPF1 mutants, including C126S (deficient UPF2 binding), R854A (deficient helicase activity), G506R/G508E (deficient ATPase/helicase activity), and S1084A/S1089A/S1100A/S1107A (4SA, lacking four phosphorylation sites). None of these NMD-deficient UPF1 mutants could rescue neuronal survival (new **Fig. 5**), strongly suggesting that the neuroprotection by UPF1 requires its NMD functions.

3. Figure 3: The authors indicated that DPRs increased the sequestration of UPF1 in the stress granule, which is the reason why NMD is inhibited. However, UPF1 was still observed in the cytoplasmic region (not stress granule) even by DPRs expression (Fig. 3a) and I am wondering whether those UPF1 is sufficient or lack for NMD. To address this, the authors need to develop and use UPF1 mutants that cannot go into the stress granule by DPRs but are functional in NMD. I do not understand the sentence "recruitment of UPF1 to stress granule is not eh cause of NMD inhibition" because knockout of G3BP did not represent NMD inhibition.

We apologize for any ambiguity in our writing. Our conclusion is that although poly(GR) and poly(PR) cause UPF1 concentration in stress granules, this is NOT the cause of NMD inhibition. Our conclusion is based on two analyses: 1) While stress granules were dramatically reduced in

G3BP1/2 knockout cells (**Fig. 3c**), DPR-induced increase in NMD target abundance is similar between WT and DKO cells (**Fig. 3d**). *ii*) Treating cells with ISRIB, an inhibitor of integrated stress response, significantly reduced polyPR-induced stress granule formation (new **Supplementary Fig. 8a**). However, it did not affect polyPR-induced NMD target accumulation (new **Supplementary Fig. 8b**). We are open to any suggestion to further improve the clarity in our interpretation.

4. Figure 4: The authors need to provide the cell fractionation was complete.

We thank this reviewer for pointing out this omission. We have added the qPCR data of U1 snRNA and GAPDH to **Fig. 4a**, confirming the effectiveness of cell fractionation.

REVIEWERS' COMMENTS:

Reviewer #1 (Remarks to the Author):

Response to revisions of "C9orf72 Arginine-Rich Dipeptide Repeats Inhibit UPF1-Mediated RNA Decay via Translational Repression"

The authors thoughtfully successfully responded to the comments we provided. They perform multiple new and clever experiments to address our concerns regarding known translational repression caused by the arginine-rich DPRs. Specifically, they perform a careful titration of cycloheximide to determine the extent that translational repression contributes to their observed stabilization of NMD substrates. Furthermore, their use of many functional mutants of UPF1 helps supports their assertion that UPF1 OE is rescuing their observed phenotypes through NMD. Finally, their use of ISRIB as an orthogonal method to reduce stress granules provides strong evidence that UPF1 rescue occurs independent of its recruitment to stress granules.

One remaining flaw within the text is that they call caffeine an NMD inhibitor, as it reduces phosphorylation of UPF1. This is technically true, but the authors should note within the text that caffeine is a very broad inhibitor of protein kinases, and its effect is not specific to inhibiting SMG1. Indeed, there are at least two more specific NMD inhibitors (NMDi and SMG1i, of which SMG1i is far more potent and specific) that could be used to reduce off-target effects on their observations. Their current observations should still hold true, but the authors must change their wording, and consider using more specific (and potent) inhibitors for future experiments.

Reviewer #2 (Remarks to the Author):

The authors have extensively revised the manuscript and appropriately addressed my concerns. The revised manuscript is much improved. Nevertheless, my major concern about this study still remains how R-DPRs block NMD. The author asserted the reason of NMD inhibition by R-DPRs is translation inhibition. Indeed, PR20 reduced the NMD targets like CHX did (Figure 4) and dominant negative UPF1 mutants decreased PR20-induced toxicity (Figure 5), suggesting that UPF1 is involved in DPR-inducing mechanisms through translation repression. This reviewer still does not recognize the detail links between translation inhibition by DPR and UPF1 function (NMD). In other words, how the authors exclude the possibility that other events in translation repression by DPR do not work on NMD: for example, translation repression should downregulate the expressions of many NMD factors. For more than two decades, many studies have provided the significant evidences of translation effects on NMD by showing, for instance, SURF complex, PYM function, eIF2alpha phosphorylation, removal of UPF1 on transcripts so on. Furthermore, translation arrest by arginine-rich dipeptide has been reported before. To reach the quality of Nature communications, this reviewer would like to ask the authors the strong and

detail novel discovery connection between NMD and DPR, not simple conclusion that translation repression by DPR blocks NMD.

Please find below our response to the referees' comments.

Reviewer #1 (Remarks to the Author):

Response to revisions of “C9orf72 Arginine-Rich Dipeptide Repeats Inhibit UPF1-Mediated RNA Decay via Translational Repression”

The authors thoughtfully successfully responded to the comments we provided. They perform multiple new and clever experiments to address our concerns regarding known translational repression caused by the arginine-rich DPRs. Specifically, they perform a careful titration of cycloheximide to determine the extent that translational repression contributes to their observed stabilization of NMD substrates. Furthermore, their use of many functional mutants of UPF1 helps supports their assertion that UPF1 OE is rescuing their observed phenotypes through NMD. Finally, their use of ISRIB as an orthogonal method to reduce stress granules provides strong evidence that UPF1 rescue occurs independent of its recruitment to stress granules.

One remaining flaw within the text is that they call caffeine an NMD inhibitor, as it reduces phosphorylation of UPF1. This is technically true, but the authors should note within the text that caffeine is a very broad inhibitor of protein kinases, and its effect is not specific to inhibiting SMG1. Indeed, there are at least two more specific NMD inhibitors (NMDi and SMG1i, of which SMG1i is far more potent and specific) that could be used to reduce off-target effects on their observations. Their current observations should still hold true, but the authors must change their wording, and consider using more specific (and potent) inhibitors for future experiments.

We are glad that this referee felt that his/her comments have been appropriately addressed. We agree that SMG1i may be more potent and specific than caffeine, and we plan to use it for future investigations. We have revised the text as requested.

Reviewer #2 (Remarks to the Author):

The authors have extensively revised the manuscript and appropriately addressed my concerns. The revised manuscript is much improved. Nevertheless, my major concern about this study still remains how R-DPRs block NMD. The author asserted the reason of NMD inhibition by R-DPRs is translation inhibition. Indeed, PR20 reduced the NMD targets like CHX did (Figure 4) and dominant negative UPF1 mutants decreased PR20-induced toxicity (Figure 5), suggesting that UPF1 is involved in DPR-inducing mechanisms through translation repression. This reviewer still does not recognize the detail links between translation inhibition by DPR and UPF1 function (NMD). In other words, how the authors exclude the possibility that other events in translation repression by DPR do not work on NMD: for example, translation repression should downregulate the expressions of many NMD factors. For more than two decades, many studies have provided the significant evidences of translation effects on NMD by showing, for instance, SURF complex, PYM function, eIF2alpha phosphorylation, removal of

UPF1 on transcripts so on. Furthermore, translation arrest by arginine-rich dipetide has been reported before. To reach the quality of Nature communications, this reviewer would like to ask the authors the strong and detail novel discovery connection between NMD and DPR, not simple conclusion that translation repression by DPR blocks NMD.

We are glad that this referee felt that his/her concerns have been appropriately addressed. As the referee pointed out, many previous studies have established how translation arrest may inhibit NMD, which is outside the scope of our study. The previous model to explain the effect of R-DPRs on NMD was focused on the relocalization of UPF1 to stress granules. We not only conclusively showed that stress granule formation is independent of NMD inhibition, but went on to show that, at least for the several tested NMD targets, translation repression could explain most if not all the effect of R-DPR on NMD. We believe these results have uncovered sufficient details of the mechanism by which R-DPRs lead to transcriptome aberrations through the inhibition of translation-dependent RNA surveillance.